# A Systemic Approach to Investigate the Gaps between Distribution System Operators Need and Technology Developers' Perception—A Case Study of an Intelligent Low-Voltage Grid Management System with Storage

**Alemu Moges Belay** [1,*]**, Sanket Puranik** [1]**, Francisco Díaz-González** [2] **and Heidi Tuiskula** [1]

1 Smart Innovation Norway AS, Håkon Melbergs vei 16, 1783 Halden, Norway; sanket.puranik@smartinnovationnorway.com (S.P.); heidi.tuiskula@smartinnovationnorway.com (H.T.)

2 Centre D'innovació Tecnològica en Convertidors Estàtics i Accionaments (CITCEA-UPC), Department of Electrical Engineering, Universitat Politecnica de Catalunya ETS d'Enginyeria Industrial de Barcelona, C. Avinguda Diagonal, 647, Pl. 2, 08028 Barcelona, Spain; francisco.diaz-gonzalez@upc.edu

\* Correspondence: alemu.belay@smartinnovationnorway.com

**Abstract:** The purpose of the paper is to introduce a new bi-directional approach to assess the gap between the customer needs and technology developers' perception on the value propositions of innovations which includes storages. The paper used two methods; the first comprehensive sense and respond analysis investigated technology developers' perceptions using the value propositions defined under the EU-funded H2020 RESOLVD project. The second method focused on customers and collected a survey which covered challenges, value propositions and preparedness to adopt new technology. The H2020 RESOLVD project has developed an intelligent low-voltage grid management system with storage. The results from the sense and respond analysis showed that most of the value propositions aligned with the responses from the broader survey which are needed within five years (e.g., improved power quality of grid, fault detection, reduced technical loss). However, the cybersecurity perception differed between developers and distribution system operators (DSOs). The customer survey highlighted that certain value propositions of technological solutions are needed more urgently than others, and therefore, technology developers should prioritize these in further developments. Regarding the use of flexibility to manage the LV grid, unclear regulations were expressed as a key barrier, thereby affecting business feasibility around battery storage.

**Keywords:** distribution system operator; low voltage grid; power electronic device; sense and respond; smart grid; storage

## 1. Introduction

The European commission has been provided public funding for enormous sustainable energy research projects to speed up renewable transition using new and emerging clean energy technologies. This is with the aims of reaching the EU renewable energy targets set for 2030 and to achieve climate neutrality by 2050. However, the UN sustainable development goal (SDG7) report, released just before the COVID-19 pandemic, clearly indicated that efforts are needed to scale-up and accelerate the share of renewable energy (RE). To facilitate and overcome the challenges of sustainable energy transition, different approaches have been proposed, focusing on technology, customers/end-users, infrastructure, the need for stakeholders' engagement, etc. For example, [1] proposed a multi-speed energy transition and discussed the opportunities and challenges of energy security by analyzing divergent energy security perceptions. The authors of [2] also investigated the challenges of renewable energy development and the critical factors which hinder renewable energy (RE) development and transition, considering social acceptance besides another eight factors.

Despite enormous efforts to overcome the challenges and facilitate the energy transition, the world seems to be behind the target set by the UN SDG and the EU. In this regard, the recent UN report showed that the energy-efficiency improvement rate fell short by approximately 3%, affordable and reliable energy is still critical in some sectors, many are not electrified, and that the share of renewables in total consumption is 17%. This clearly shows that much faster growth is required to meet long-term renewable and climate goals. To speed up the energy transition, advanced technology and modernization of the grid infrastructure and management system are key.

Smart grid technology has received special attention for the last decade since there are market drivers, such as digitalization of the distribution grid and optimization of the network operations [3]. However, while adopting advanced technologies and products, it is important to engage customers in the whole process of the value chain and understand customer challenges, e.g., electricity prices [4]. Involving and interacting with customers enables technology developers to satisfy customers' needs and have a high market share in the future. Two-way communication between the developers and end-users will reward active and loyal customers. According to [5], responding to customers' needs and updating them about what the technology offers will enable the scaling of user-centered value into big business. This active engagement of customers will help developers to understand the changes in customer expectations and experiences in a continuous manner and develop mitigation measures proactively.

In fact, the relevance of and a need for a better method/approach for customer engagement has long been discussed in other sectors, including energy and automotive sectors. For example, [6] investigated internal and external sources using the voice of customer (VoC) concept and revealed that users/customers who possess the needs and problems are major sources for innovative ideas. Another concept which has been discussed in literature is co-creation, which involves the customer from the start to the end. Co-creation with users results in more meaningful innovation [7,8] and is an established design practice [9,10]. Studies have shown that successful new products and services involve users in various developmental phases [11,12]. There are also other design-thinking methodologies, such as agile methods, to engage customers.

Most of these methods are used for assessing customer satisfaction using a survey, interview or analyzing historical data using statistical methods [13,14]. However, most research on customer engagement and technology development and adoption is one-directional; either the assessments are completed by customers (e.g., surveys) or vice versa. Moreover, most of such research is conducted with commercially available products and services with validated values, unlike the low/medium Technology Readiness Level (TRL) research projects. In this regard, this paper combined both the sense and respond (S&R) approach together with a broader survey as a validation of the value propositions. Indeed, the S&R approach has been applied in other sectors, such as manufacturing, processes, services, etc. [15–17]. S&R was proposed as one of the methodologies due to the fact that it enables analysis with smaller sample sizes of respondents. Moreover, it is easily adoptable and applicable in various sectors, as long as respondents are able to rate the four attributes, i.e., expectations of the value propositions, experiences, have knowledge about competitors and understand the general trend/direction of technology (see Table 1). S&R needs a clear and understandable attribute (factor to be considered). In this paper, value propositions which were well-designed and formulated based on the RESOLVD technological solutions were implemented.

**Table 1.** Sample questionnaire for S&R method.

| RESOLVD Value Proposition (VP) | Expectations (before the Pilot Results) | Experiences (after the Pilot Results) | Compared with Competitors (Please Tick the Box below) | | | Direction of Development (Please Tick the Box below) | | |
|---|---|---|---|---|---|---|---|---|
| | (1–10) | (1–10) | Worse | Same | Better | Worse | Same | Better |
| VP 1 | [ ] | [ ] | ☐ | ☐ | ☐ | ☐ | ☐ | ☐ |
| VP 2 | [ ] | [ ] | ☐ | ☐ | ☐ | ☐ | ☐ | ☐ |
| • | [ ] | [ ] | ☐ | ☐ | ☐ | ☐ | ☐ | ☐ |

Expectation = what is the expectation of the attributes; experience = what is the experience of the attribute; compared with competitors = compare experienced value to the values of all other targets; direction of development = direction of the experienced values of the sample during the last three years.

This paper considered the EU-co-funded low/medium Technology Readiness Level (TRL) research project (H2020 RESOLVD) to create a bi-directional communication between technology developers and DSOs for better value creation and to make proactive decisions. The aim of the H2020 RESOLVD project was to develop and validate, in field, a set of software and hardware technologies revolving around contributing to better management of a rural distribution grid towards increasing the hosting capacity for renewable generation. In particular, the technologies targeted an optimal power flow within the grid, exploiting advanced network monitoring devices and an innovative, hybridized battery-based energy storage system. The overall RESOLVD objectives and detailed architecture of the technology, developed together with the needs, involvement and expectations of stakeholders, are presented in [18]. In a broader sense, the paper paves a way towards open innovation by leveraging internal and external sources of ideas and taking those ideas (value propositions) to the market [19,20].

### 1.1. Objectives and Research Questions

The objectives of this research were twofold: (i) investigate the gap between customer needs and technology developers' perceptions of these needs and (ii) validate the value propositions offered by the new LV grid solution developed in the project. The solution under investigation was a comprehensive package of smart grid technologies, developed to improve the operation and management of the electricity distribution network. The paper introduced a sense and respond (S&R) methodology to identify the gap between the technology developers' perceptions and the target customer needs (DSOs). The S&R method was further complemented by a customer survey to validate the relevance and urgency of the value propositions. The objectives can be broken down into four specific research questions:

1. What are the challenges, status and practices of LV grid management?
2. How relevant and urgently needed are the value propositions of the RESOLVD project?
3. How prepared are the DSOs to deploy the new technology?
4. Is there any regulation barrier/challenge which hinders RES integration at the LV grid level?

### 1.2. Organization of the Paper

The paper starts by briefly discussing next generation smart grid technology (Section 2) and introduces a sense and response method, considering the value propositions of the project, and targeting the technology developer with specific questions constitutes Section 3. This was done by targeting the technology developers with specific questions to investigate their perception of the value propositions of the project. A broader survey consisted of four parts, targeting the DSOs, and an analysis will also be presented in Section 4. Finally, and before the concluding remarks, the paper discusses the reflections and highlights the

gaps from the sense and response results (technology developers) and the broader survey (DSOs) in Section 4.

The steps through which the major research procedure was organized is presented in Figure 1 below.

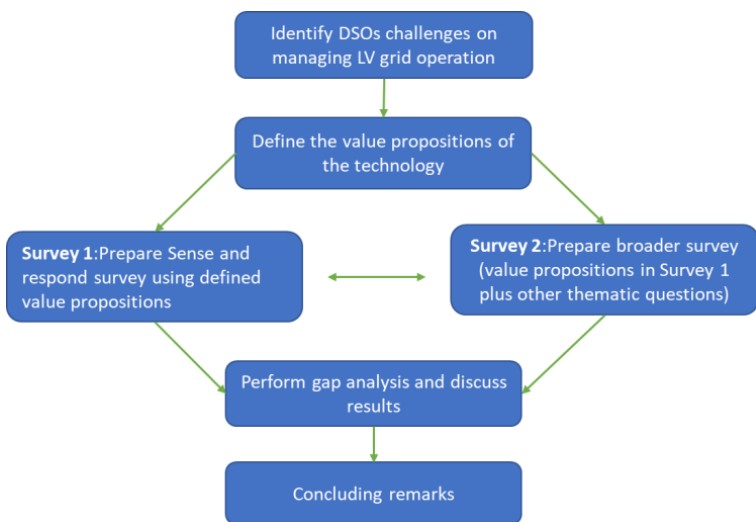

**Figure 1.** Organization of the research.

## 2. Smart Grid Project Overview in the EU

Smart grids are a response to the need for modernizing existing grid infrastructure so as to increase efficiency, maintain safety and reliability and achieve large-scale integration of RES [21,22]. Smart grid is an umbrella term which covers multiple technologies, concepts and application domains. The authors of [22–24] provide a comprehensive overview on the status of smart grid technologies. All of these studies clearly have shown that future grids increasingly are going to be digitized under the foundation of ICT infrastructure and will use large amounts of data to perform better decision making.

To track the progress of smart grids in the EU, the Joint Research Centre (JRC) periodically publishes a report under the smart grid project outlook (hereafter, SGP report), the latest of which was published in 2017 [25]. In this report, the JRC categorized smart grid projects under six domains, of which smart network management is the domain which had most of the projects and, thus, investments. Smart network management comprises applications such as tools for network observability, advanced sensors on network equipment to identify anomalies and communicate with nearby devices when a fault occurs, tools for self-healing and tools for reactive power control and energy storage, to name a few.

As much of DERs (including RES) are expected to be connected to a distribution grid, the distribution system operators (DSOs) have been identified as key players in the transition towards a clean energy system [26]. DSOs shall be crucial to sustainable energy transition and, thus, have a central role in the revised Directive on Common Rules for the internal market for electricity, which is part of the Clean Energy for all Europeans legislative package, issued in 2019. DSOs in the EU have acknowledged this role, and to equip themselves, they have been investing in smart grid projects, as reflected in the SGP report [27,28]. Furthermore, according to [29], DSOs are expected to invest heavily so as to establish complete system innovations to their traditional businesses.

Based upon the SGP report, there are more R&D projects compared to demonstration projects. This suggests that more R&D is needed before solutions can reach market. It should be noted that demonstration projects are more capital-intensive than R&D projects. This makes demonstration projects more difficult to implement from a financial point of view and require more confidence in the technology. About 60% of investments in demonstration projects come from private investments, while for R&D projects, they account for

only 40%. The remaining investments in R&D projects come from either EU or national funding schemes, which, ultimately, is public money. This observation has also been seconded by [30]. Thus, if technological innovations from R&D projects are not adopted by the market, it results in a loss of public money. Developing new grid technologies and processes involves high uncertainties concerning, for instance, performance, replicability, scalability and consumer response [31,32]. These uncertainties can negatively affect investment decisions and can be reduced by decreasing the gap between technology developers' perceptions and customer needs in the early phases of R&D.

## 3. Methodology

### 3.1. Two Types of Surveys

Most of the EU low and medium Technology Readiness Level (TRL) projects are not mature enough for commercialization in the market, and they need a set of validation criteria by listening to the voices of end-users/target customers (DSOs) and incorporating the feedback of technology developers. In relation to this connection and as part of a continuous learning process, H2020 RESOLVD designed and employed two types of surveys, targeting the feedback from both the technology developers and the DSOs:

1. Sense and respond: the target participants are technology developers of the H2020 RESOLVD project i.e., consortium members of the project from five EU countries.
2. A broader survey with a specific target customer of the DSO, which is highly bounded by regulations.

The feedback collected from the two sides were expected to minimize the gap and enhance the market potential of the H2020 RESOLVD outcomes by better understanding the functional and market requirements of the proposed solutions. The feedback would have been able to give insight on what and how to prioritize the critical value propositions if the customers had specific interests in the selected solutions which the H2020 RESOLVD offered. The two types of surveys which targeted the technology developers and DSOs, with their main contents, are presented in Figure 2.

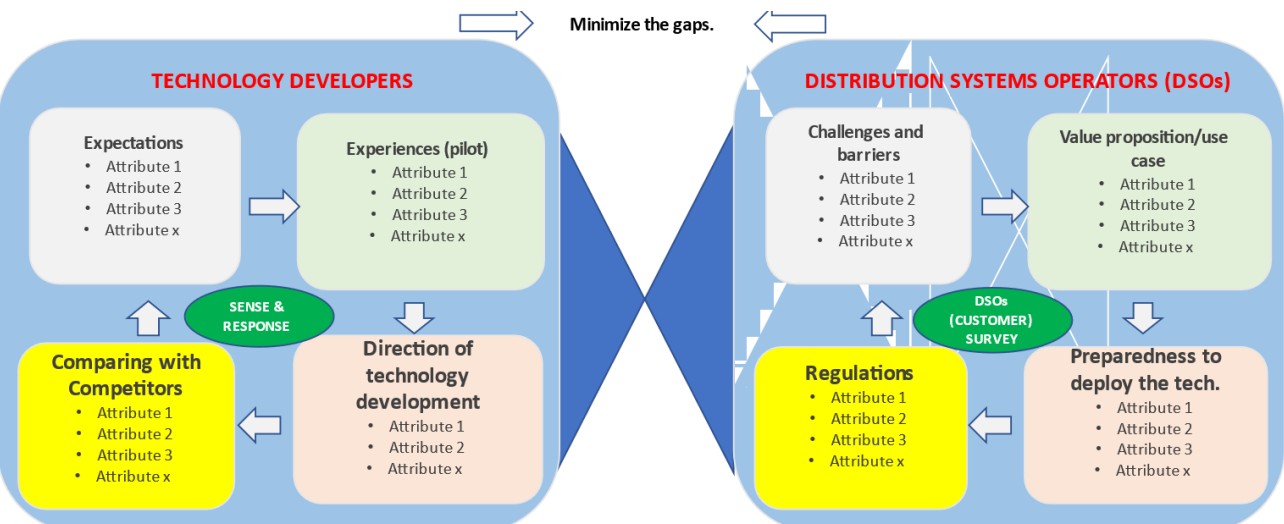

**Figure 2.** The two types of surveys for technology developers and DSOs.

The two methods focused on the overall RESOLVD solution, comprising of value propositions which were gained from the four technologies developed. All value propositions were incorporated into the S&R questionnaires and the broader survey. Because the main goal of the project was to overcome current and future challenges of the DSOs in a global context, the paper also considered other important aspects (barriers, preparedness, regulations), as presented in Figure 2 of the broader survey, with the content explained

in Section 3.3. The results and discussion are presented in Section 5. As the objectives of this research focused on the technology outcomes of the project and seeing the gaps between technology developers' perceptions and DSOs' needs, more emphasis was placed on validating the value propositions and incorporating them into both the surveys.

### 3.2. Sense and Respond (S&R) Approach (Target: Technology Developers)

In general, the sense and respond approach created a two-way conversation with the market and used these conversations to create value by understanding the customer needs [5]. In line with this approach, the H2020 RESOLVD conversations (bi-directional communication) between technology developers and the target customers (DSOs) would serve to validate how the proposed H2020 RESOLVD solutions were relevant by first sensing and responding accordingly.

In H2020 RESOLVD, we introduced the sense and respond analytical tool, which combined the expectations of technology developers from the project value propositions, experiences of the proposed solution, comparisons to the competitor/available solution and investigations of the general technology development (see sample questionnaire in Table 1). The results from the analysis then provided a holistic view on identifying the critical value proposition based on the respondents' expertise knowledge (technology developers). This ultimately helped technology developers to prioritize between different value propositions of the complete RESOLVD solution and, thereby, adjust their business strategy and joint exploitation pathway. The analytical sense and respond method was adopted and used for analysis purposes with the governing formula [15,16].

The Critical Factor Index (CFI) was calculated according to [16] and presented as follows:

$$\text{CFI} = \frac{\text{Standard deviation of expectations} * \text{Standard devation of experiences}}{\text{Importance index} * \text{Gap index} * \text{Direction of tech. development index}}, \quad (1)$$

$$\text{Gap index} = \left| \frac{\text{Average of the experiences} - \text{Average of expectations}}{10} - 1 \right|, \quad (2)$$

$$\text{Direction of development index} = \left| \frac{\text{Dir. of devt.(better)} - \text{Dir. of devt.(worse)}}{10} - 1 \right|, \quad (3)$$

The gap index considers the change of respondents' experiences and expectations (before and after the deployment of the technology in the pilot and measured KPIs) (Equation (2))

The direction of development (Dir. of devt.) index indicates the subjective judgment of respondents based on their experiences of the past, current and expectation of future trends of technology evolvement in relation to specified attributes (Equation (3)). The respondents' rated if the proposed attribute was better, worse or same as business as usual, as presented in Table 1.

By incorporating the sample correction error [15], the Scaled CFI was calculated as:

$$\text{Scaled CFI} = \frac{\text{Standard deviation of expectations} * \text{Standard devation of experiences}}{\text{Gap index} * \text{Direction of development index} * \text{Importance index}} - \frac{s-1}{2 * \ln(2) * n}, \quad (4)$$

where '$s$' is the minimum number of samples required, and '$n$' is the number of total samples (responses) analyzed; CFI is the critical factor index without sample error correction.

The importance index was considered by averaging the given experiences scores (see Table 1) after piloting. The experience was scored based on the results of the project KPIs from the pilot compared to the business-as-usual scenario (BAU). The competitor index reflects a relative performance of the attribute, considered and rated by technology developers, based on the available competitor's technology.

*3.3. Broader Survey (Target: Distribution System Operators (DSOs))*

Following the feedback from the technology developers through the S&R method, a broader survey was conducted to explore the market potential and assess the customers' acceptance of the proposed H2020 RESOLVD solutions. In the meantime, the survey intended to validate if the DSO feedback was a valuable source of new ideas and served to gauge and prioritize if further improvements were needed. This survey was organized into and covered four parts:

- Part 1: Current status and practices of the LV grid management system: In this section, sets of guiding questions were designed to understand the status, challenges, barriers and future needs of the LV grid management system. These questions gathered DSOs' feedback on how resource intensive the LV grid operation is and how critical it is to detect all types of losses and got an overview on the impact of increased RES and activation of flexibility while managing an LV grid.
- Part 2: RESOLVD value propositions to overcome DSO challenges: The feedback in these parts was directly linked to the list of value propositions which H2020 RESOLVD offered. This was designed point-by-point in response to all of the value propositions and KPIs listed. These included the improvement in observability; continuity of supply to customers, thanks to energy storage capability; reduction in distribution power losses; improved power quality; increase in the grid hosting capacity for renewables, among others. A complete list of the value propositions of RESOLVD is presented in Section 4.
- Part 3: DSOs' preparedness to adopt new technologies: To analyze the future market potential of RESOLVD solutions and their deployment, a set of questions were designed to be responded proactively regarding the preparedness of the DSOs in adopting new technologies which would enable them to manage the LV grids intelligently. These questions were designed to assess the position and readiness of the DSOs to implement new technology, hear the customers' (DSOs) voice about if the required standards were in place, implement protocols and interfaces to foster interoperability between legacy systems, etc. and further investigate if there were any constraining factors to the use of flexibility in grid management.
- Part 4: Regulatory aspects to tackle challenges of RES integration at the LV grid level Complying with regulations and legal requirements is eminent while managing smart grids. Therefore, it is clear that looking at the challenges and barriers linked with regulations and legal aspects of the LV grid management is necessary. Therefore, the survey covered questions regarding if the DSOs had encountered any specific national regulatory, legal and other constraining factors. Another important aspect incorporated into the survey was to investigate if the e-regulation and e-directive of the EU commission (under the clean energy package, 2019) was clear enough to foster the use of flexibility assets. Furthermore, the need for updated regulations with respect to the potential business models and approaches for successful adoption of new technologies, such as the H2020 RESOLVD solutions, were assessed.

## 4. RESOLVD Solution for Smart Grids

Renewable penetration levered by Efficient Low-Voltage Distribution grids (RESOLVD) is a low/medium Technology Readiness Level (TRL) project, funded under the European Union's Horizon 2020 scheme. The project has developed a complete solution to support DSOs to improve the efficiency and hosting capacity of distribution networks in a context of highly distributed renewable generation, by introducing flexibility and control into the low voltage grid [33].

The overall project's cognitive process can be understood from all the technologies functioning in an integrated way to improve LV grid performance. Phasor measurement units (PMU) and smart gateways measure power-related attributes to instantiate real-time grid observability. In the project, these elements were installed at key locations between primary and secondary substations of the distribution grid. All the data from the PMU and

smart gateways, together with the smart meters, went to a low-voltage distribution decision support tool (LVD-DST), which then provided actionable insights to the grid operator. The LVD-DST could also control certain grid elements directly to perform needed actions in an automated manner. A power electronic device (PED), developed in the project, is a grid element which gets signals from the LVD-DST to provide required grid services through multiple storage devices. The PED comprises of hardware and software components connecting multiple storage devices, and it contributed to achieving most of the value propositions (described later in this section). Through PED, it was also possible to isolate certain sections of the grid and operate them in an island mode. All of the new monitoring and control elements were integrated into a legacy system using a customized enterprise service bus (ESB) and data management platform. In addition to these technologies, a risk-based threat-modeling approach for smart grid technologies has been developed to ensure cybersecurity. Figure 3 shows the different layers of the DSO operation where the RESOLVD technologies fit.

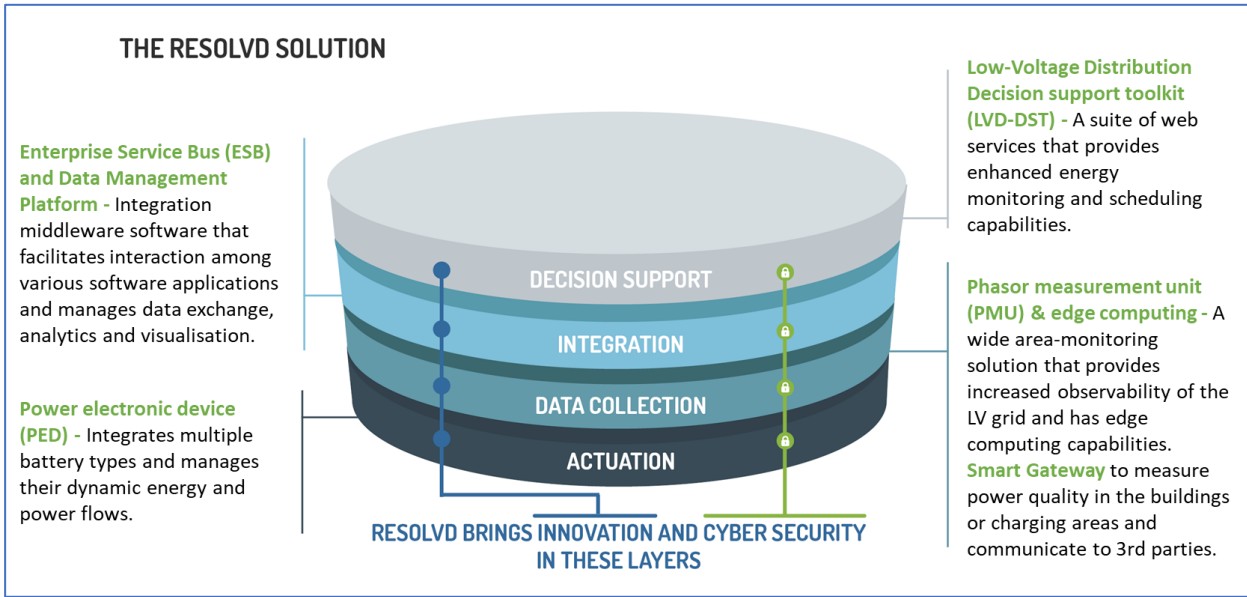

**Figure 3.** The H2020 RESOLVD solution comprised of technologies falling under four layer of DSO operation, as shown in the figure.

The technologies in more detail are:

- Low-voltage distribution decision support tool (LVD-DST): A suite of web services that provides enhanced energy monitoring and scheduling capabilities. It has three key functionalities: (1) Demand and generation forecasting: uses smart meter data for day-ahead forecasts and prediction of congestion and voltage problems. (2) Fault detection and isolation provides enhanced monitoring of the grid based on multivariate statistics to automatically detect faults and other abnormalities in a statistical sense. (3) Optimal grid operation scheduling calculates the optimal grid operation schedules of active elements (switchgear/storage) in the grid to prevent critical events, reduce energy exchange at substation level and peak shaving.
- Power electronics device (PED): This is an innovative power electronics cabinet that permits the integration and synergistic exploitation of various battery types. In the H2020 RESOLVD project, the PED integrated a lead-acid battery pack and a lithium-ion one. The PED was operated so that the main performance of each battery type could be maximized, i.e., the relatively low degradation of the lithium-ion battery in comparison to the lead-acid one under stringent power requirements and the possibility of allocating part of the total required energy storage capability to a low-cost lead-acid battery, thus reducing the cost of the overall solution. The PED was

connected to the LV side of a transformer of a secondary substation in a rural area, which served as the pilot location for the H2020 RESOLVD project. Key functionalities performed by the PED were providing flexibility to the low-voltage grid, ensuring security of supply in case of grid failure, active and reactive power control at the secondary substation level for voltage and transformer loading management, current harmonics compensation and unbalance correction among the three phases of the distribution grid. A detailed description of the PED can be found in [34,35].

- Phasor management unit (PMU) and edge computing: A wide area monitoring solution that provides increased observability of the LV grid and has edge computing capabilities. The PMU-based solution allows for real-time monitoring, protection and control of the power grid. This allows the system to operate closer to the margin and react to network disturbances.
- Smart Gateway: A gateway to measure power quality in the buildings or charging areas and communicate to third parties. The power quality monitoring capability allows for the measurement of meter information and for it to be translated to services. It allows for communication with third party services and triggers actions.
- Enterprise Service Bus (ESB) and Data Management Platform: This is integration middleware software that facilitates interaction among various software applications that are relevant to a DSO. It also manages data exchange, analytics and visualization. Key functionalities include integration with legacy systems of the DSO, customization to smart grid systems, data exchange, analytics and visualization of key performance indicators.

The integrated RESOLVD solution could help in achieving eight specific value propositions. These value propositions also corresponded to the use cases designed and tested in the project. The proposed and achieved value propositions of the project were:

1. Improved capability to detect and interrupt unintentional uncontrolled islanding
2. Improved continuity of supply under intentional controlled islanding
3. Enhanced capability to prevent congestion and over/under voltage issues
4. Improved voltage control ability (through local reactive power injection)
5. Improved power quality (through power electronics)
6. Reduced technical losses (through power electronics and local energy storage/local generation)
7. Improved fault detection (observability) and critical event prevention
8. Improved cybersecurity

## 5. Result and Discussion

### 5.1. Result from S&R

The sense and response (S&R) method was implemented with the intention of incorporating and understanding the overall perceptions of technology developers on the H2020 RESOLVD solutions, and the project values were proposed. The method considered four dimensions to assess the relevance of the value propositions defined. These included the expectations of the developers for each proposition (before the piloting), the experiences after the KPIs were measured and quantified at the demonstration pilot, comparisons with other competitive technologies and the partners' general overview on the direction of technology development under the scope of smart grid management [18]. The overall summary of the S&R, with the scaled CFI values, which comprised all components of the attributes, such as experience, expectations, gap index, competitor index and direction of technology development, and results from the Equations (1)–(4) are presented in Table 2.

**Table 2.** Sense and response attributes with a scaled CFI.

| No. | Average of Expectation | Standard Deviation of Expectation | Average of Experiences | Standard Deviation of Experiences | Gap Index | Direction of Development Index | Competitor Index | Importance Index | CFI with Sample Error Correction (Scaled CFI) |
|---|---|---|---|---|---|---|---|---|---|
| 1 | 6.83 | 1.17 | 6.83 | 1.72 | 1.0000 | 0.5000 | 1.0000 | 0.6830 | 6.013 |
| 2 | 6.83 | 1.60 | 7.67 | 1.83 | 0.9160 | 0.5000 | 1.1667 | 0.6830 | 9.4804 |
| 3 | 7.67 | 1.75 | 7.5 | 1.76 | 1.0170 | 0.5000 | 1.3333 | 0.7670 | 8.0172 |
| 4 | 7.83 | 1.72 | 8.33 | 1.75 | 0.9500 | 0.5000 | 1.3333 | 0.7830 | 8.2132 |
| 5 | 7.60 | 1.82 | 7.60 | 1.82 | 1.0000 | 0.5000 | 1.3333 | 0.7600 | 8.837 |
| 6 | 7.60 | 2.30 | 8.00 | 2.12 | 0.9600 | 0.8333 | 1.3333 | 0.7600 | 8.1399 |
| 7 | 7.00 | 2.12 | 7.60 | 2.34 | 0.9400 | 0.8333 | 1.5000 | 0.7000 | 9.1673 |
| 8 | 7.00 | 2.19 | 7.16 | 1.94 | 0.9840 | 0.8333 | 1.1667 | 0.7000 | 7.5219 |

By aggregating these four dimensions and applying the critical factor index (CFI) analysis presented in the methodology section, the top three critical value attributes (see Figure 4 highlighted in green) were identified, i.e., fault detection and critical event forecasting, voltage control through local reactive power injection and reduced technical losses through power electronics and local storage/local generation. With almost the same level of CFI values, the second top three value propositions were identified (see Table 2 CFI with sample error correction and Figure 3 highlighted in light green), which were the power management in intentional controlled island mode, prevention of congestion and over/under voltage issues through local storage utilization and grid reconfiguration and improved power quality through power electronics.

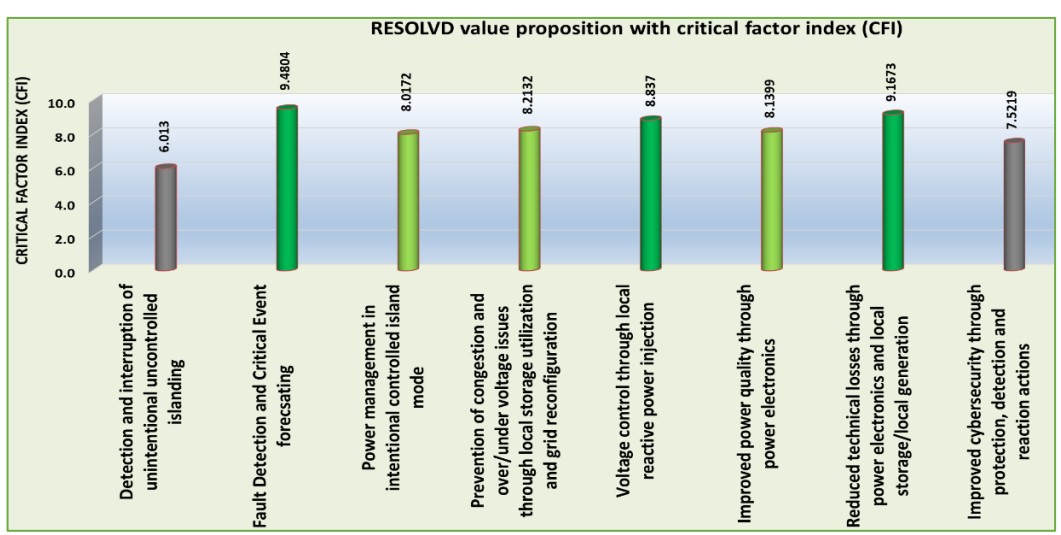

**Figure 4.** Critical factor index of the RESOLVD value propositions based on the sense and respond analysis.

The overall technology developers' perceptions of the H2020 RESOLVD solutions, with respect to each value proposition and critical factor index, is presented in Table 2.

To overcome the LV grid challenges based on their criticality, the pareto analysis, which is known as the 80/20 principle, was used to systematically prioritize the propositions based on the CFI values. The pareto analysis showed the 80% of the challenges could be traced using the top six value propositions (Figure 5 green and light green), followed by improved cybersecurity through protection/detection/reaction actions and detection and interruption of unintentional uncontrolled islanding (see Figure 4 highlighted in gray color). Using similar contents of the critical factor analysis, the paper further implemented the pareto analysis as a methodology to justify how the decision makers could systematically compare the value propositions from technology developers' perspectives against the broader survey targeting the DSOs. The pareto analysis was done to prioritize the value propositions and solve the linked challenges in a step-by-step manner. The results from the CFI analysis and pareto analysis varied slightly and were very close. These close values

could be due to the smaller sample size of technology developers' feedback. In this regard, further research would be needed with a large number of technology developers to avoid the inconclusiveness and for a better visualization of the results.

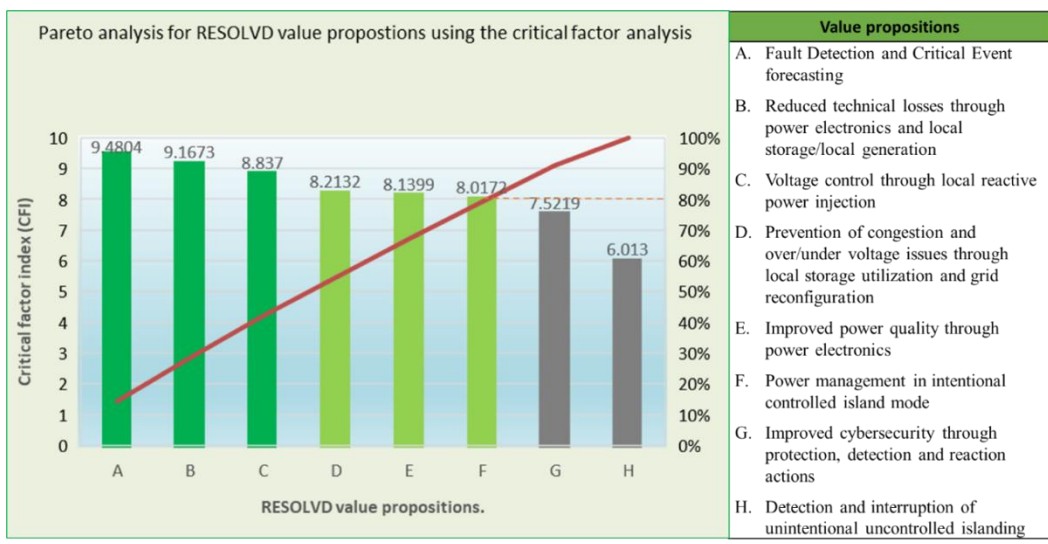

**Figure 5.** Pareto analysis for the RESOLVD value propositions using the critical factor analysis (CFI).

One of the value propositions, i.e., detection of non-technical losses (such as frauds) through advanced sensors (WAMS) and data analytics, was assessed as an indirect value from the whole H2020 RESOLVD solution for the DSOs. It served only for internal retribution and renumeration calculations. That is why only the DSOs and partners with a very close interaction to the pilot responded (rated) to this attribute, while others faced difficulties in doing so. However, a few responses were not completed because not all partners had the same level of understanding of each detailed technology. Therefore, for the purposes of analysis, a business-as-usual (BAU) scenario was considered.

### 5.2. Results from the Customer Survey

A customer survey was performed to gather customer feedback concerning the value propositions offered by the H2020 RESOLVD solutions. This section provides key results of the survey, which was answered by 21 DSOs from 10 countries (nine from the EU and one from India). The survey was designed based on the RESOLVD technology, which would be applied within the DSOs which faced multifold challenges in managing LV grids with increasing penetration of distributed generation. For sense and response, the paper considered five technology developers with one pilot DSO. As discussed in the Methodology section, the sense and response methodology required a minimum of three responses for analysis purposes, and for this reason, the proposed method was best suited to this research. Obviously, like other statistical analyses, the greater the number of responses, the better the results and the more minimal the inconclusiveness. As a result of this, similar research with a large number of technology developers would be needed. However, it is also important to understand the challenges of getting large responses in a highly regulated sector, like energy.

#### 5.2.1. Part 1: Current DSO Status and Practices

From Figure 6, it is clear that LV grid operation is a resource-intensive business, constituting a significant share of DSOs' yearly expenditure. A total of 81% of respondents either strongly agreed or agreed that loss detection (both technical and non-technical) was still critical for DSO business. This is in line with the Council of European Energy Regulators' (CEER's) latest report on losses, where many DSOs in Europe have recorded losses above

5% [36]. Most DSOs were in favor of the opinion that increased renewable penetration was likely to cause problems with traditional operations of the grid. There has been clear evidence from various studies on the impact of high renewable energy penetration, particularly on distribution grids, including a report by the European commission [28]. The report also proposed a broader role of DSOs in the future to tackle the challenges associated with renewable energy source integration. Still, 14% of the responding DSOs thought that growing renewable energy was not going to cause any problems for their businesses. The reason could have been that they had already invested in a grid upgrade and/or were confident with their current grid capacity. However, the reason was not clear and needs further investigation.

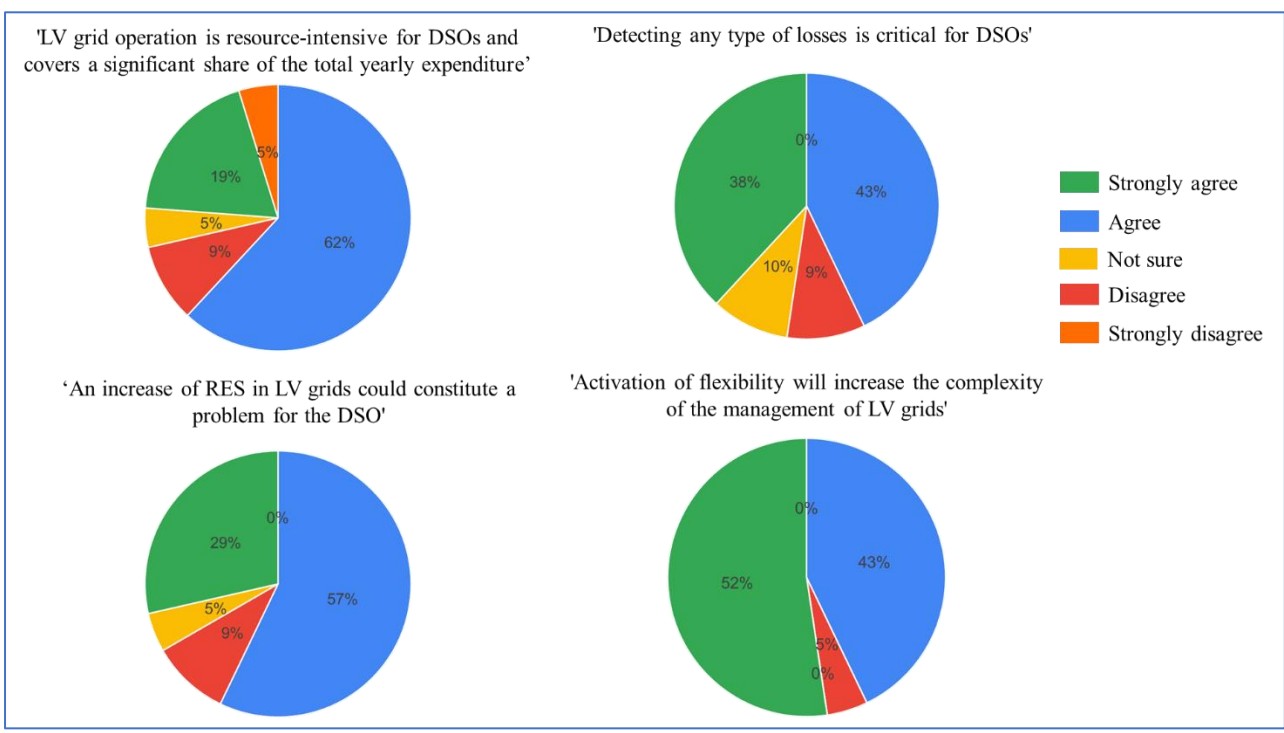

**Figure 6.** Customers' opinions on four topics related to LV grid operation.

Regarding observability, 48% of DSOs were still limited to secondary substation monitoring. Only 9% of DSOs were using real-time data from smart meters at the customer level to know the grid status, while 24% were making use of smart meter data at the aggregated level for observing the grid (see Figure 7a). About 19% of DSOs were using smart meter data only for billing purposes. Figure 7b shows that more than 50% of DSOs still relied on customers to detect faults in the grid. Studies have shown that smart meter data can be used for detecting faults in the distribution network faster than traditional approaches [1]. The results showed that DSOs were missing out on getting this added benefit, even though most of the respondents were from countries with a high smart meter penetration. There was a clear gap between having access to data and the utilization of data for benefits. To take advantage of smart meter data, DSOs require specific tools, which RESOLVD has proposed to deliver through its LVD-DST, thereby filling the gap in market.

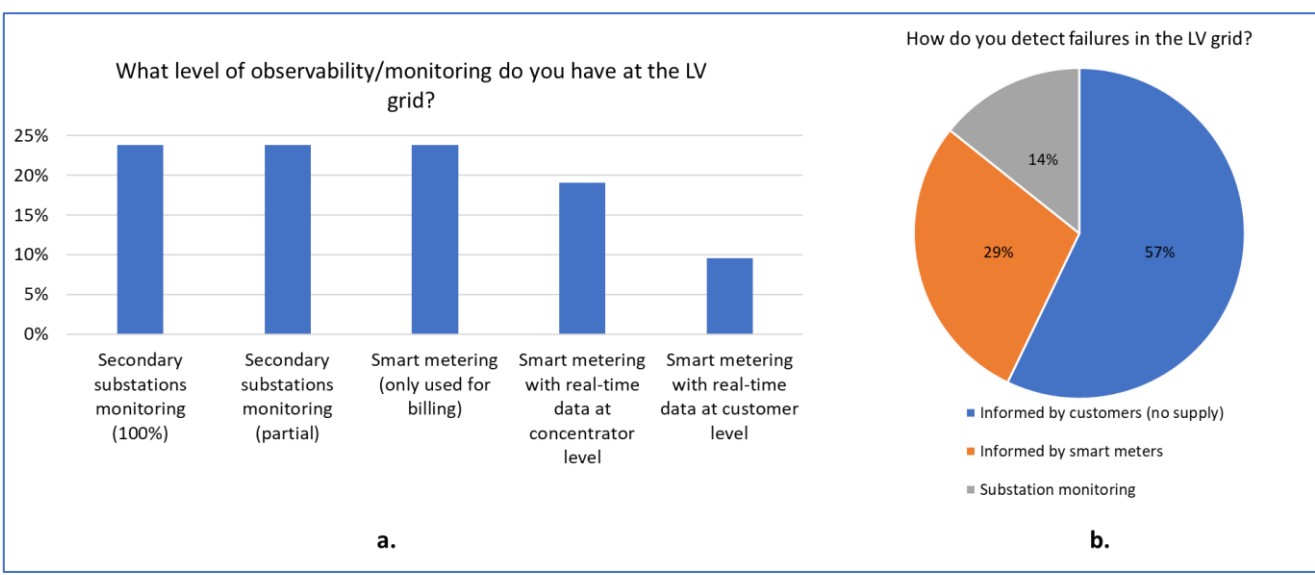

**Figure 7.** Customers' responses to (**a**) the level of observability in their operation area and (**b**) the method of fault detection.

Most of the DSOs considered cybersecurity issues important, which were mostly addressed internally (by 86% of DSOs). A small proportion of DSOs outsourced these issues to external experts, while 9% did not address cybersecurity issues at all. This has a profound implication on the business strategy for cybersecurity methodology being developed in the project. About 76% of respondents either strongly agreed or agreed that DSOs could benefit from flexibility markets, thus reflecting a positive attitude towards them.

5.2.2. Part 2: RESOLVD Value Propositions

Part 2 comprised two types of questions. The first one was a list of high-level questions to grasp and validate the overall objectives of the project. These included the need to improve observability of the LV grid, the ability to integrate multiple LV grid components (such as the legacy system and upcoming IoT technologies), exploit flexibility of distributed energy storage and improved decision support tools (forecasting, scheduling and event detection). In this regard, the results from the respondents showed that there was an urgent need (within five years) to improve the decision support tools and observability of the LV grid (see Figure 8). Meanwhile, the flexibility and ability to integrate multiple LV grid components would be relevant in 5–10 years. Indeed, this depends on the availability of infrastructure and resources at the respondents' premises and their plan to reach a high penetration of DERs. It is interesting to note also, from the pareto analysis presented previously in Figure 5, that the possibility of managing distributed energy storage was seen as relevant and prioritized, but that more than 60% of the respondents' (DSOs) left this question as being most relevant in 5–10 years time. This could have been associated with the still manageable share of distributed renewable generation throughout the territory; thus, an urgent need of energy storage was mostly associated with the pace of the decarbonization of the grids.

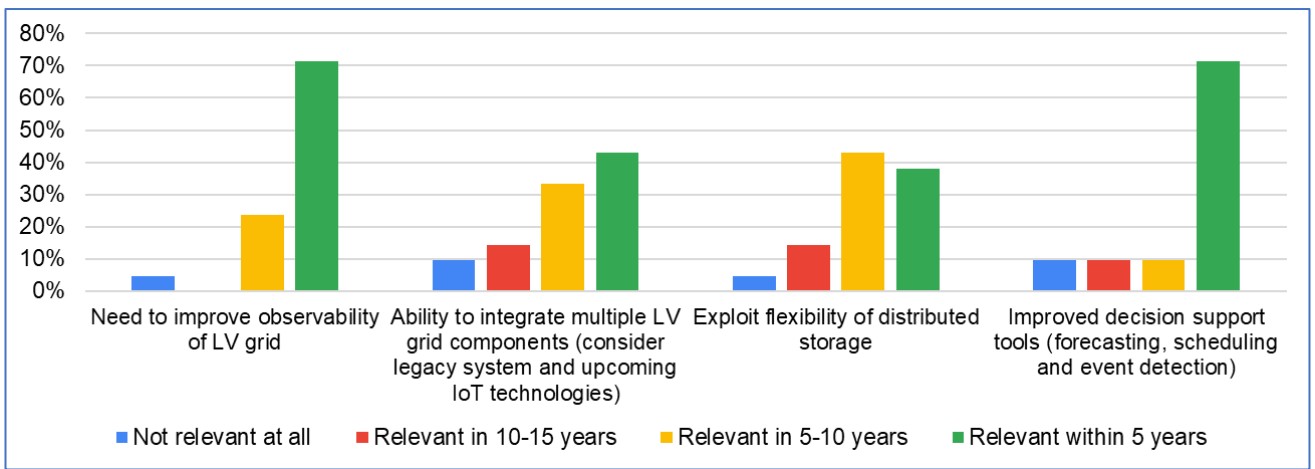

**Figure 8.** High-level DSO demands pertaining to the RESOLVD objectives.

The second part of the questions solely focused on the H2020 RESOLVD value propositions and were designed with the intention to validate and create synergy between the H2020 RESOLVD technology developers and the target customers (DSOs). For this reason, the same value propositions were analyzed using S&R, targeting the technology developers.

Figure 9 shows the responses on the relevancy and urgency of the H2020 RESOLVD value propositions. About 70% of the respondents believed that three of the value propositions (i.e., improved capability to prevent congestion and over/under voltage issues, improved power quality and improved fault detection and critical event prevention) would be relevant in the coming five years, with importance shown by technology developers in the S&R results as well. Cybersecurity issues were also ranked to be relevant in the near future but had a lower CFI in the S&R results. One reason for this could have been that most customers (DSOs) have their own internal cybersecurity service, as this was revealed on the survey conducted, and other technology developers have limited knowledge of cybersecurity and have different perceptions. According to the responses, value propositions related to islanding mode were likely to be more relevant in a 10–15-year timeframe. For the local reactive power injection capability and reducing technical losses value propositions, the DSOs were mostly split between less than five years and a 10–15-year timeframe. A probable reason for this split could have been due to differences in the renewable penetration rate in customers' operating regions.

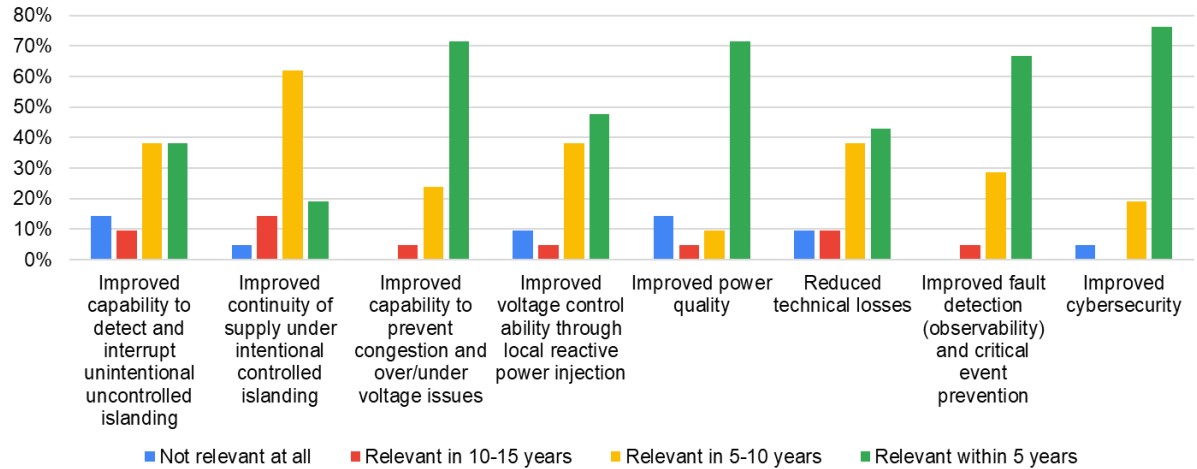

**Figure 9.** Customer responses to the urgency and relevancy of the RESOLVD value propositions.

The DSOs' responses showed that the RESOLVD value propositions were relevant in overcoming the existing and future challenges at the LV grid level. However, the urgency of the value proposed varied among the respondents. In general, the responses from the customer survey (DSOs) and S&R analysis (mostly technology developers) complemented each other on most of the value propositions. This indicates synergy between technology developers' expectations pertaining to the RESOLVD solution and market needs. However, the results from both methods had small variations on a couple of the value propositions. An interesting observation is that from the DSOs perspective, cybersecurity was the first most rated proposition within a five-year timeline, which slightly differed from the technology developers' views with a low CFI. Furthermore, 67% of the customers would have liked to get a combination of few value propositions, rather than a complete package (see Figure 8). This shows that different customers had different needs, and thus, the solution should be modular enough to be customizable for different customers.

It is also worth noting, from Figure 9, that many of the H2020 RESOLVD value propositions, such as improve the continuity of supply to customers, improve the capability to prevent congestion and over/under voltage, improve the voltage control ability through local reactive power injection and improve power quality, were all tightly related to the possibility of managing energy storage systems in grids. Thus, in contrast to the previous analysis depicted in Figure 8 on the DSO needs, mostly stressing grid observability, new software for decision support and interoperability as the most relevant needs in the short term, the analysis in Figure 7 suggests that to actually solve grid operational aspects, energy storage technologies and associated power electronics are key assets to be exploited, following the software tools and observability capabilities previously mentioned.

### 5.2.3. Part 3: DSOs' Preparedness

A lack of necessary infrastructure was the only reason cited for a lack of DSO preparedness. This highlights the fact that some DSOs would need additional grid upgrades before they could benefit from RESOLVD solution (see Figure 10). The majority of DSOs have encountered regulatory/legal barriers while trying to utilize flexibility for grid management.

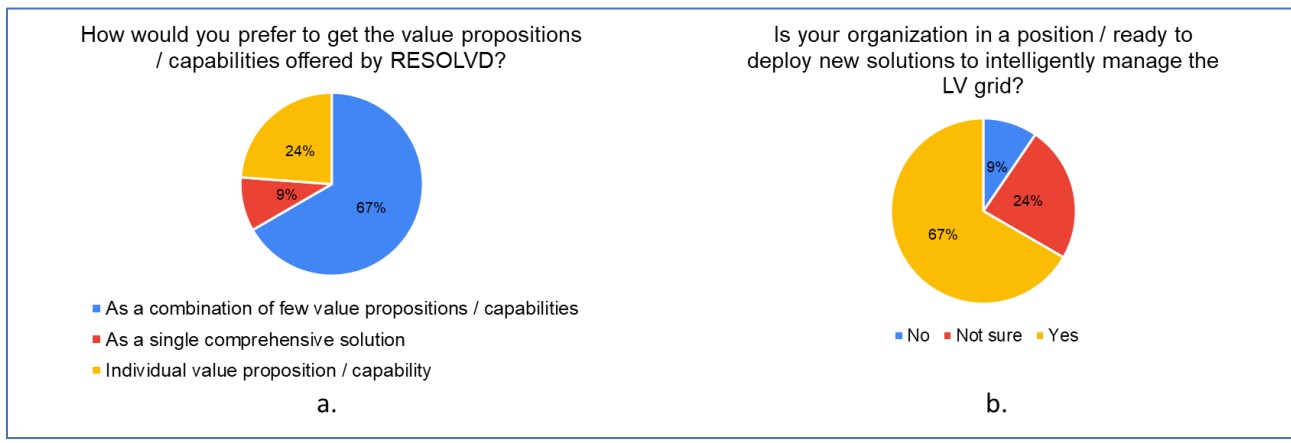

**Figure 10.** Delivery of the RESOLVD solution and DSO preparedness.

### 5.2.4. Part 4: Regulatory Aspects

About 43% of DSOs thought that existing e-regulations and e-directives from the EU commission were partially clear to foster use of flexibility by DSOs. This lack of clarity was also identified in the report by the EU commission, which stated that the provisions in the policies were too general due to separation of DSOs in two simple categories [28]. About 95% of DSOs agreed that regulations needed to be updated to allow more innovative business models to be adopted by DSOs, while the remaining 5% strongly disagreed on

this. Regarding future remuneration approaches for DSOs, views were split; 76% thought that a TOTEX (Total Expenditure)-based approach was appropriate, while 24% did not know what approach would be best in the future.

The responses from the DSOs in connection to the regulatory aspects clearly showed that a lack of a regulatory framework is a challenge. Typically, with respect to the flexibility of services and markets, the respondents highlighted that the current law did not yet allow flexibility in the network, and there were no prices or conditions for the DSOs for RES integration at LV grids. In this regard, DSOs were not allowed to add flexibility items, e.g., BESS, demand response or other devices/services as part of regulatory asset value. which was the basis for regulated revenue. Only classical DSO grid items were accepted, which did not promote large scale deployment of optimal solutions. Other important aspects which were emphasized by respondents were the disallowance of battery ownership (storage), only partial regulatory frameworks for a flexibility market, missing incentives for DSO transformation (e.g., a trough network usage fee) and missing allowed network fee for reverse direction of electricity flow. In this regard, for example in Germany, operation of energy storage by the DSO is not regulated, and using the smart meter solution to control producing or loads was not released by the IT security authority (BSI) until now. A new law is expected to be passed in 2021/22 about controlling consumer loads in LV grids.

### 5.3. Gap Analysis and Key Findings

Although most of the RESOLVD value propositions were pretty aligned with the customers, the research identified some gaps which could be filled through better engagement between the developer and the DSOs.

- A high importance was given to cybersecurity by DSOs, which was not reflected as strongly in the S&R.
- Not all value propositions were urgently needed.
- Value propositions related to uncontrolled and controlled islands were relevant but not urgent. According to DSOs, these value propositions were expected to become urgent in the span of 5–10 years. Therefore, for these value propositions, the technology developers have more time to mature technology than others.
- For technologies needed withing five years, the technology developers should expedite their innovation process to be market-ready in time.

The main takeaway and contribution of this paper was the introduction of the sense and response approach in the energy domain combined with the broader survey. This provided a better understanding of the customer's (DSOs) needs and the technology developers' perceptions on their technology value propositions. This two-way communication and interaction minimized the gap and facilitated rapid learning by sensing the most critical challenges and responding accordingly. At a higher level of decision making, e.g., the EU, this could elicit in-depth insight into which direction of research is relevant and should be prioritized to overcome the current and future challenges of the DSOs as well as the right technology based on the need and urgency.

### 6. Conclusions

Engaging and understanding customer needs is crucial to successful market pickup of newly developed technology and innovation. Smart grid R&D projects are capital-intensive and are often funded through public money with the intention to solve the challenges of target customers (pubic and commercial entities). Thus, it is important that customer needs are aligned with technology developers' perceptions about the market as early as possible in development process. This paper provided a method to identify gaps between technology developers' perceptions and customer needs in the low Technology Readiness Level (TRL) smart grid project.

The method was applied on technologies developed in the H2020 RESOLVD project. The results showed that the challenges which DSOs face under high renewable penetration were in line with technology developers' perceptions. However, the need and the urgency

of technology varied, which could be associated with the local context of DSOs (like regulations and renewable penetration level). The untapped potential of using data from smart meters has been identified, which could be leveraged as an important selling point for the decision support toolkit developed in the project. The results from the broader survey showed that three value propositions, namely improved fault detection and critical event prevention, improved voltage control ability and reduced technical losses, are the ones which will be needed urgently by DSOs within five years. The S&R analysis which focused on the technology developers showed that respondents also rated these value propositions highly. This indicates that technology developers should prioritize maturing solutions so as to fulfil urgently needed value propositions. The paper showed that cybersecurity was rated as relevant and urgent by DSOs but had lower critical factor index (CFI) values among technology developers. Most of the DSOs showed interest and were prepared for implementing the RESOLVD solution, but the ones which were not indicated a lack of infrastructure as the reason for not being prepared. Most of the DSOs identified unclear regulation as a barrier to use flexibility in grid management. This has important business implications for the success of the power electronic device in the market. The methodology proposed in this paper has provided key insights for technology developers to strategize further technology development in line with DSO needs.

**Author Contributions:** A.M.B. and S.P.: conceptualization, methodology, formal analysis, draft preparation, review and editing; H.T.: review and editing, supervision; F.D.-G.: draft preparation, formal analysis. All authors have read and agreed to the published version of the manuscript.

**Funding:** The RESOLVD project has received funding from the European Union's Horizon 2020 research and innovation program under grant agreement No. 773715. The information and views set out in this study are those of the author(s) and do not necessarily reflect the official opinion of the European Union. Neither the European Union institutions and bodies nor any person acting on their behalf may be held responsible for the use which may be made of the information contained therein.

**Institutional Review Board Statement:** Not applicable.

**Informed Consent Statement:** Not applicable.

**Acknowledgments:** Thanks to Richa Parsai for supporting the design of the customer survey.

**Conflicts of Interest:** The authors declare no conflict of interest.

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
