# Peer review of "A Systemic Approach to Investigate the Gaps between Distribution System Operators Need and Technology Developers’ Perception—A Case Study of an Intelligent Low-Voltage Grid Management System with Storage"

_applsci, doi:10.3390/app11125348_

Round 1
Reviewer 1 Report
This article tries to analyze the differences between the needs of distribution system operators and the perception of technology developers about innovations. It uses the innovation proposals established in the RESOLVD project (funded by the EU, H2020) and customer surveys. In the reviewer's opinion the subject matter of the article is doubtful for inclusion in the journal Applied Science. In case of acceptance, the following aspects should be reviewed:
- The Pareto analysis performed (Figure 4) does not seem relevant since it is repetitive with respect to the content of Figure 3. It is also inconclusive for the identification of priority propositions given their distribution.
- The target audiences of the customer survey conducted (section 5.2) in 10 countries does not seem compatible with the technology developers according to the 5 EU consortium members participating in the H2020 RESOLVD project (section. 3.1). This aspect should be clarified to give consistency to the comparative analysis performed in the article.
- The sentence on line 346 seems misplaced as it is after table 2 and explanation of figure 3.
- The reference to figure 4 in line 419 is not correct.
- Some inconsistencies in the text should be revised (e.g., legend table 2, beginning of sentence in line 513, etc).
Author Response
Dear Reviewer,
We really do thank you for your valuable and constructive comments. We presented our response in a point by point manner in table 1. We attached your comments (a reviewer 1) together with other reviewers feedbacks.
Best regards
Alemu

Reviewer 2 Report
The paper briefly presents the results of RESOLVD project, and although the findings can be considered as valuable, the presentation is poor.
The references are not by MDPI standards, please check the Instructions for authors.
Also, some references are mentioned in the paper, but are not included in the list of references (i.e. Belay et al 2013)
Equations are not in accordance with Instructions for authors - please do the complete overhaul.
Overall, paper need to be improved, and if it wasn't for some valuable data, it would be for reject decision.
Author Response
Dear Reviewer,
We really do thank you for your valuable and constructive comments. We presented our response in a point by point manner in table 1. We attached your comments (a reviewer 2) together with other reviewers feedbacks.
Best regards
Alemu

Reviewer 3 Report
The paper presents an approach used to assess the gap between the technology development for low voltage networks with distributed generation and the needs of DSOs - customers to whom this technology is addressed, based on the RESOLVD project.
The topic is interesting, up-to-date and relevant for the development of distribution networks and their transition to smart grids. The paper is organized well and written in rather a clear way, however in the reviewer opinion some improvement should be made.
Comments
- The presented method has been designed and applied to validate the relevance of the solutions offered in the RESOLVD project. However, similar approach could be used to in relation to any technologies and customers to identify gaps between the customer needs and developers perceptions. Could authors comment on to what extend the presented methodology can be applied in other cases? Are there any general rules or conditions that should be fulfilled to have reliable conclusions?
- Solutions offered by the RESOLVD include different hardware and software technologies. Were the value propositions matched to a specific technology, or were they the same in all S&R questionnaires? How many responses and from which stakeholders were collected?
- There are some explanation missing regarding S&R approach. What is the CFI index? The index is introduced on row 207 but explanation appears only on row 337. How the importance index in CFI is determined? What is a competitor index? What are criteria for experience judgement?
- The literature Diaz et al. 2019 (row 290) is not included in References section.
- RESOLVD project was presented in the International Conference on Electrical Power Quality and Utilisation in 2020: Samper et al. “RESOLVD: ICT services and energy storage for increasing renewable hosting capacity in LV”, and issues such as needs and expectations of the involved stakeholders were discussed. Making a reference would be nice.
Author Response
Dear Reviewer,
We really do thank you for your valuable and constructive comments. We presented our response in a point by point manner in table 1. We attached your comments (a reviewer 3) together with other reviewers feedbacks.
Best regards
Alemu

Round 2
Reviewer 2 Report
1. The reference fix is wrong. Please read carefully the Instruction for Authors:
- "...numbered in order of appearance and indicated by a numeral or numerals in square brackets—e.g., [1] or [2,3], or [4–6]."
The first reference that appears is [21], then [30], [1], [7], etc...
If you use some reference software, please use corresponding Style format. In case you do not use reference software, please reorganize the references manually, taking care that the content corresponds to cited reference.
2. Keyword "Technology" is too general, please be more specific. Also, keywords need to me organized alphabetically.
3. Please address the numeral TRL level in lines 94 and 180.
4. Although the equations are now numbered, for what reason has their shape changed?
Additionally, the equations should be called up and explained in the text of the paper.
5. The real strength and scientific contribution of the paper is in Chapter 4, which should be emphasized more strongly and should describe the cognitive process by which the solutions were reached. The authors may already have these parts in the technical reports or in the project application and can use this information here.
6. Chapter 5 is great, congratulations to authors!
Author Response
Dear Reviewer2,
Dear Reviewer 2- Round 2
Many thanks for your positive and constructive comments for the betterment of the paper. The paper is now improved, and we inserted the cognition processes to show the knowledge we/the project aims to contribute (a separate paragraph inserted to show how the RESOLVD solution works. In addition, other comments are fixed accordingly. The formula is also as previous version for better visualization. We referred the equations in the text. Our point-by-point reply for your second-round comment is presented in table 1 attached.
Best regards,
Alemu

Round 3
Reviewer 2 Report
The authors did all corrections as asked and the paper is improved in many ways.